# Multi-language Diversity Benefits Autoformalization

**Albert Q. Jiang**
University of Cambridge
qj213@cam.ac.uk

**Wenda Li**
University of Edinburgh
wenda.li@ed.ac.uk

**Mateja Jamnik**
University of Cambridge
mateja.jamnik@cl.cam.ac.uk

## Abstract

Autoformalization is the task of translating natural language materials into machine-verifiable formalisations. Progress in autoformalization research is hindered by the lack of a sizeable dataset consisting of informal-formal pairs expressing the same essence. Existing methods tend to circumvent this challenge by manually curating small corpora or using few-shot learning with large language models. But these methods suffer from data scarcity and formal language acquisition difficulty. In this work, we create `MMA`, a large, flexible, multi-language, and multi-domain dataset of informal-formal pairs, by using a language model to translate in the reverse direction, that is, from formal mathematical statements into corresponding informal ones. Experiments show that language models fine-tuned on `MMA` can produce up to $29-31\%$ of statements acceptable with minimal corrections on the `miniF2F` and `ProofNet` benchmarks, up from $0\%$ with the base model. We demonstrate that fine-tuning on multi-language formal data results in more capable autoformalization models even on single-language tasks.

## 1 Introduction

Formal mathematics refers to mathematical content that is represented in a formal language that can be mechanically checked by a computer. Practitioners express mathematics in formal languages integrated into proof assistants like HOL Light [Harrison, 1996], Isabelle [Paulson, 1994], Coq [Barras et al., 1999], and Lean [de Moura et al., 2015]. *Autoformalization* is the task of translating natural language materials into verifiable formalisations. An ideal autoformalization engine can reduce the excessive cost for modern mathematical results to be verified [Ball, 2012, Scholze and Stix, 2018]. It opens up the vast amount of mathematics expressed in natural language to automated reasoning research fields that rely on formal languages, like automated theorem proving [Wu et al., 2022].

The hope of automatically translating informal mathematics into formally verifiable content is as old as formal mathematics [Whitehead and Russell, 1925–1927]. Only very recently, the breakthroughs in neural networks and Neural Machine Translation (NMT) enabled autoformalization to be learned [Wang et al., 2020, Wu et al., 2022, Jiang et al., 2023b]. NMT methods typically require a large *parallel dataset*, that is, a dataset consisting of pairs of sequences expressing the same meaning in both the source and the target language. The most challenging part of autoformalization research is constructing such a parallel dataset in a natural and a formal language, satisfying two conditions simultaneously: (1) the natural language component is close to how mathematics is actually written; and (2) the number of datapoints is large enough for the data-hungry machine learning methods. This is hard, because manually translating informal mathematical content into a formal language is only doable by highly trained experts in both mathematics and computer science, hence costly.

In this work, we address the lack of a parallel dataset by leveraging a state-of-the-art Large Language Model (LLM), GPT-4 [OpenAI, 2023]: we used it to translate the two largest formal corpora, Archive of Formal Proofs in the language of Isabelle, and mathlib4 in the language of Lean4, into natural language. This process was enabled by the key observations that informalisation is much easier than formalisation, and a powerful LLM can produce diverse natural language outputs. As a result, we

38th Conference on Neural Information Processing Systems (NeurIPS 2024).

Table 1: Example parallel pairs from MMA.

| Isabelle statement | GPT-4 informalisation |
|---|---|
| `lemma eint_minus_le:`
`  assumes "(b::eint) < c"`
`  shows "c - b > 0"` | The lemma named "eint_minus_le" assumes that an extended integer "b" is less than another extended integer "c". It then shows that the result of "c" subtracted by "b" is greater than zero. |
| `lemma closed_superdiagonal:`
`  "closed {(x,y) | x y.  x ≥ (y::`
`  ('a::{linorder_topology}))}"` | The set of all pairs of elements (x, y) such that x is greater than or equal to y, is a closed set in the context of a linearly ordered topology. |

| Lean4 statement | GPT-4 informalisation |
|---|---|
| `theorem norm_eq_one_of_pow_eq_one`
$\{\zeta : \mathbb{C}\}\ \{n : \mathbb{N}\}\ (\mathtt{h} : \zeta^n = 1)\ (\mathtt{hn} : n \neq 0)$:
$\parallel \zeta \parallel = 1 :=$ | For a complex number $\zeta$ and a natural number n, if $\zeta$ to the power of n equals 1 and n is not equal to 0, then the norm of $\zeta$ is equal to 1. |
| `theorem mul_dvd_mul_iff_left`
$\{a\ b\ c : \mathbb{N}\}\ (\mathtt{ha} : 0 < a) : a * b \mid a * c$
$\leftrightarrow b \mid c :=$ | For any three natural numbers a, b, and c, where a is greater than 0, a times b divides a times c if and only if b divides c. |

created a parallel dataset of 332K informal-formal pairs, which we refer to as the MMA (Multi-language Mathematical Autoformalization) dataset. To the best of our knowledge, this is the first dataset of natural-formal language aligned data with more than one formal language. The only similar work was that of Azerbayev et al. [2023], which has only one formal language (Lean3) and is 4x smaller than our dataset. Four examples of MMA are shown in Table 1.

We fine-tuned two open-source LLMs, LLaMA-33B [Touvron et al., 2023] and Mistral 7B [Jiang et al., 2023a], on MMA to generate corresponding formal expressions given the informal ones. The trained model was then evaluated on two autoformalization benchmarks, `miniF2F` and `ProofNet`. Manual inspection of 50 problems for each model from each benchmark showed that after fine-tuning, the models could produce $29 - 31\%$ of formal statements on the benchmarks that require no or minimal correction, whereas the raw model produced $0\%$. We also fine-tuned two identical models on the Isabelle and the Lean4 components of MMA separately for the same number of steps. Their autoformalization performances are significantly weaker than the model trained on multi-language data, demonstrating that parallel data containing multiple formal languages is crucial for autoformalization training.

**Contributions:**

- We informalise all formal statements from the Archive of Formal Proofs and mathlib4, creating MMA, a dataset of informal-formal pairs. This is the first natural-formal language aligned dataset containing multiple formal languages.
- We train the first language models that can autoformalize to multiple languages in the zero-shot setting, and manually evaluate them on two autoformalization benchmarks.
- We verify that: (1) language models trained on MMA acquire strong autoformalization abilities; and (2) language models trained on MMA have greater autoformalization performance than those trained on single-language partitions of it with the same computational budget.
- We release the fine-tuned models for inference. We also release the MMA dataset for people to train their autoformalization models on, and to enrich MMA with more domains and languages.

Improving autoformalization ability of models has the potential of translating copious digital repositories of informal human knowledge into formal languages of reasoning tools, and thus presents an opportunity to formally verify human informal arguments and solutions. High quality datasets such as MMA and autoformalization models like ours pave the way towards this goal[1].

---

[1]The MMA dataset and the fine-tuned models are available from the official repository: MMA.

## 2  Related Work

**Autoformalization Datasets.** Wang et al. [2018, 2020] manually aligned a small parallel dataset and generated a larger parallel dataset with a rule-based informalisation tool [Bancerek, 2006] from Mizar to LaTeX. Manual alignment is almost as expensive as formalising mathematics anew. Moreover, unlike generative neural informalisation tools (e.g., GPT4), symbolic informalisation tools such as Naproche [Cramer et al., 2009] result in natural language content that lacks the inherent diversity and flexibility in expression: they are rigid and not natural-language-like. Finally, symbolic informalisation tools are hard to design and implement. They also differ a lot for different formal languages, hence the approach is not scalable for multiple formal languages.

Wu et al. [2022] sought to eliminate altogether the need for a parallel dataset by leveraging the in-context learning ability of LLMs: they provided a couple of parallel examples, and asked the LLMs to find a formal counterpart for the informal problem (limited to high-school algebra or number theory). This approach is very effective when the test domain is limited. But when there are many test domains, finding the correct parallel examples becomes difficult: the LLM invents syntactically incorrect segments when it does not know the formal syntax for certain concepts [Wu et al., 2022, Case Study 3]. Liu et al. [2023] and Huang et al. [2024] both utilised autoformalization to create aligned informal-formal pairs of data that are verified either manually or mechanically (for proofs), but did not perform large-scale synthesis of corresponding informal-formal theorem statements. Li et al. [2024] provides a more detailed survey on autoformalization datasets. In summary, there is no existing method, like the one we propose here, that is scalable both in terms of formal languages and mathematical domains.

**Back-translation.** In natural language machine translation literature, the quality of translation heavily depends on the quality of the parallel data between two languages. However, for all but a few language pairs (e.g., `en-fr`), such parallel data is rare and hard to curate [Guzmán et al., 2019]. Back-translation is one of the most effective methods to improve translation quality [Sennrich et al., 2016, Artetxe et al., 2018] in this setting, which is similar to ours. Back-translation uses an existing target-to-source model to turn ground-truth target sequences into noisy source sequences. Then, it bootstraps a source-to-target model to reconstruct the ground-truth target from the noisy source.

Usually, the back-translation process is practised in both directions of translation, that is, from source to target and from target to source, and is iterated until convergence. When back-translation is practised in one direction only (because the model from target to source is called through an API and not trainable, for example), this process is referred to as "distilled back-translation". Azerbayev et al. [2023] used OpenAI's Codex [Chen et al., 2021] model to perform distilled back-translation to improve their own model's autoformalization capabilities. `MMA` differs from their dataset mainly in that `MMA` contains data from multiple formal languages and has four times as many datapoints.

**Language Models for Executable Programs and Reasoning.** Since OpenAI's Codex [Chen et al., 2021], multiple LLMs have been trained for code completion and infilling that stem from natural language [Yu et al., 2018, Austin et al., 2021, Fried et al., 2023]. Related is also the research on natural language mathematical and logical reasoning [Cobbe et al., 2021, Lewkowycz et al., 2022, Shi et al., 2022] that demonstrates that LLMs can comprehend mathematics and produce reasoning chains in natural language to a degree. Interestingly, distillation from larger, more capable models can effectively boost the reasoning ability of smaller models [Fu et al., 2023]. However, none of these works trained language models for the task of autoformalization, which is the gap that our work fills.

## 3  Dataset

As established above, there is no existing parallel corpus that satisfies the following crucial criteria for autoformalization model training:

1. The informal data is diverse and flexible, similar to natural mathematical communication.
2. The size is suitable for neural model training ($\geq$ 100K datapoints).

**Informalisation.** In this work, we use a powerful neural model (GPT-4) to generate informal data from existing formal libraries (informalisation) to create a high-quality parallel corpus. We argue, both analytically and empirically, that informalisation is an easier task than formalisation. Hence, our approach of leveraging the power and flexibility of language models for informalisation indeed produces a parallel corpus that satisfies both of the criteria above.

Formal languages have two vital characteristics that distinguish them from natural languages: (1) precision and (2) syntactic rigidity. By precision we mean that every piece of information must be explicitly and precisely expressed and formalised; whereas in natural language, pieces of information are often left implicit or ambiguous. For example, one may write in natural language `"Two roots of the equation` $x^2 - 3x + 2 = 0$`,` $x_1$ `and` $x_2$`, sum up to 3."` meaning the two distinct roots have a sum of 3. Expressed formally, one must also write $x_1 \neq x_2$ to make the statement provable. Hence, the information in the formal statement is always sufficient for the informal statement to be inferred, while the reverse is not always true. By syntactic rigidity of formal languages we mean that formal grammars are usually much stricter than natural grammars, permitting less choice and diversity when expressing the essence of a piece of information.

Wu et al. [2022] found that $76\%$ of 38 high-school mathematical problems informalised by OpenAI's Codex model were "more-or-less correct". Azerbayev et al. [2023] did a more comprehensive study on 371 university-level problems and discovered that the same model has a $62.3\%$ informalisation accuracy, while its formalisation accuracy is $13.4\%$. Empirically, informalisation has a much higher chance of being completely correct than formalisation.

**Curation Process.** Lean4 and Isabelle are two of the most popular proof systems for formalising mathematics, with by far the largest formal proof repositories: Isabelle's Archive of Formal Proofs (AFP) and Lean4's mathlib4. They total over 5 million lines of code as of May 2024. In this paper, we consider the languages of these two systems due to their sizes and their popularity within the mathematical community, although the curation process can be easily extended to other proof languages as well. In neural translation systems, similar languages tend to have similar performances as source or target languages [Lample and Conneau, 2019, Roziere et al., 2020]. Given this fact and cost constraints (see Section 7 for the curation cost), we only use the languages of Lean4 and Isabelle as target languages in this paper, and expect conclusions reached with them to generalise to similar proof languages. Lean4 and Isabelle cover a wide range of topics, from advanced mathematics to software, hardware, and cryptography verification. We use Portal to Isabelle [Jiang et al., 2021] to extract 244K theorem statements, and the LeanDojo [Yang et al., 2023] library to extract 88K theorem statements. Isabelle AFP articles are under either a BSD-style license (a modified 3-clause BSD license) or the GNU LGPL license. Mathlib4 is under an Apache 2.0 license. The derived informal statements fall under licenses identical to their formal counterparts.

We choose the most generally performant language model available to us, GPT-4 [OpenAI, 2023], to informalise the statements, since its ability with code and natural language is superior to that of Codex [Chen et al., 2021], which was used by previous works on autoformalization with LLMs [Wu et al., 2022, Azerbayev et al., 2023]. Existing works on informalisation [Wu et al., 2022, Azerbayev et al., 2023] typically use few-shot prompting to generate good informal statements. Our informalisation targets all available formalised content, going beyond high-school and undergraduate-level mathematical exercises. But targeting such a wide range of domains means that acquiring high-quality parallel pairs for every datapoint is challenging and expensive. Hence, instead of manually curating aligned pairs for every mathematical domain, we used an instruction prompting approach [Ouyang et al., 2022], adopting the instruction prompt below for informalisations, with the text in curly brackets replaced by the individual datapoint content:

```
Statement in natural language:
{$natural_language_statement}
Translate the statement in natural
language to {Isabelle|Lean}:
```

For all informalisations, we generated a maximum of 512 tokens from GPT-4 with greedy sampling (i.e., temperature $= 0.0$ in the OpenAI API). The responses received from this informalisation process often begin with "The lemma states that", which is mechanical and does not impact the meaning of the sentence. We remove such phrases and capitalise the remaining sentence.

**Statistics.** In Table 2 **(top)** we give the relevant statistics of our `MMA` dataset, including the number of datapoints for each library and the statement lengths in characters for each language.

**Analysis.** Since formal statements are precise and rooted in exact underlying definitions and complex contexts, the LLM informalisation process may sometimes fail to capture this precision. It might overlook or loosen crucial elements of the formal information, or introduce incorrect details (hallucination): this is a limitation of our work. To calibrate the extent of this limitation and further

Table 2: **(top)** Statistics of `MMA`. **(bottom)** Categorisation of errors in 200 `MMA` informalisations.

| | AFP | | mathlib4 | |
|---|---|---|---|---|
| Datapoints | 244238 | | 88536 | |
| Length (chars) | Informal | Isabelle | Informal | Lean4 |
| Mean | 340.0 | 166.0 | 288.5 | 107.8 |
| Median | 291 | 125 | 268 | 93 |
| Min | 95 | 7 | 98 | 21 |
| Max | 1546 | 24331 | 1258 | 989 |

| Error type | Isabelle | Lean4 |
|---|---|---|
| None | 81 | 67 |
| Hallucination | 2 | 6 |
| Misunderstanding concept | 11 | 18 |
| Incorrect assumption | 2 | 9 |
| Incorrect conclusion | 2 | 6 |
| Incorrect type | 4 | 8 |

characterise the dataset, we conducted a qualitative study on 200 statement pairs from the `MMA` dataset, that we detail below.

We randomly selected 100 Isabelle and 100 Lean4 datapoints from `MMA`, and manually examined each pair of informal-formal statements. We rated each pair along the following axes:

- Correctness (whether the informalisation is completely correct)
- Hallucination (whether the informalisation contains content not in the formal statement)
- Misunderstanding concept (taking one concept in the formal statement for a different one)
- Incorrectly translating assumption
- Incorrectly translating conclusion
- Incorrectly translating type

We found that 67 of the 100 Lean statements are informalised correctly, and 81 of the 100 Isabelle statements are informalised correctly. The overall correctness rate is 74%. We estimate the total correctness rate of the `MMA` dataset to be similar. Wu et al. [2022] found that even when only 25.3% of the autoformalization statements are completely correct, downstream theorem proving applications were still able to benefit drastically from the parallel dataset. Hence, we expect our `MMA` dataset, which is 3x more accurate, to be of great usefulness for the community.

In Table 2 **(bottom)** we present the types of errors out of the 200 randomly selected informalisations. Note that one informalisation can potentially have multiple errors. We notice that the most common mistake made by GPT-4 in informalising is "Misunderstanding concept", which happens in 14.5% (29/200) of the translations. This is either because there is an inherent ambiguity in the formal expression and the context is not enough to determine it, or that the language model is not able to determine the appropriate concept. Spotting these errors requires a significant amount of expertise in both mathematics and formal languages. Designing an automatic filter to remove incorrect informalisations seems to be highly non-trivial. We leave improving the informalising language model, such that it produces more accurate translations, for future work.

**Case Study.** We study the informalisation examples from Table 1: 3 of the 4 are correct, but when informalising the lemma "eint_minus_le", GPT-4 interprets the type "eint" to be extended integers, which are usually defined as normal integers extended with negative and positive infinities. This translation is sensible, but not entirely correct: "eint" is introduced in a theory of $p$-adic numbers to represent the codomain for the $p$-adic valuation – this means that it only extends integers with positive infinity, which serves as a maximal element in the order (i.e., the valuation of $0$). Therefore, it is important to note that while we use a state-of-the-art LLM (GPT-4) to perform the informalisations, the resulting `MMA` dataset is not perfect: rather than the ground truth, informalisations in `MMA` should be treated as *noisy approximations* of it.

## 4 Experiment

To validate that `MMA` is a useful dataset for models to gain autoformalization abilities, we train two models from the LLaMA family and the Mistral family on a series of `MMA` data partitions. We manually evaluate the resulting models on two downstream benchmarks: `miniF2F` [Zheng et al., 2022] and `ProofNet` [Azerbayev et al., 2023], consisting of high-school mathematical competition and undergraduate-level mathematical exercise problems respectively.

**Experimental Details.** We take LLaMA [Touvron et al., 2023] 33B (under the LLaMA license) and Mistral [Jiang et al., 2023a] 7B (under an Apache 2 license) as the base models, for they were the most performant open-weights model that we could fine-tune at the time of experimenting. We deliberately choose two models of different sizes and families to show that the improvement brought by `MMA` dataset is not sensitive to model size or family. For fine-tuning, we use the cross-entropy loss with the loss on the input masked out. We use the EasyLM [Geng, 2023] software framework on a TPUv4-64, with 32 megacores. We parallelise the model across 16 devices, and use a local batch size of 8 sequences, with each sequence having a maximum of 512 tokens. We use the AdamW optimiser [Loshchilov and Hutter, 2019], perform 5000 linear warmup steps with a peak learning rate of $3 \times 10^{-5}$, and then decay the learning rate with a cosine schedule for 35000 steps to $3 \times 10^{-6}$. Preliminary experiments suggest that the final checkpoints of models are the strongest ones, so we use those to represent fine-tuning runs.

**Fine-tuning Data Regimes.** We trained the models for the same number of training steps to generate formal statements given their informal counterparts, on different partitions of `MMA`: Isabelle + Lean4; Isabelle only; Lean4 only. For each datapoint, we used a prompt format identical to the one in Section 3 but with reversed input/output languages, and instructed the model to translate the statement in natural language to Isabelle or Lean accordingly. There are 88K informal-formal pairs of Lean4 data in one epoch of `MMA`, while for Isabelle there are 244K, 3 times as many. To reflect these proportions fairly, we fine-tuned the jointly trained model for 3.3 epochs, the Isabelle only model was fine-tuned for 4.4 epochs, and the Lean4 only model was fine-tuned for 13.2 epochs.

It is possible that the ratio between data of the two formal languages influences the models' performances and a sweep of experiments over this ratio is potentially valuable. However, since fine-tuning the LLaMA model costs $2885 by TPU pricing, we are constrained by our budget and unable to perform this sweep.

## 5 Results

In this section, we analyse the performance of the trained models and their formalisation of realistic mathematical problems from high-school competitions and undergraduate-level courses.

**Loss and Accuracy.** In Figure 1, we plot the loss and the token accuracy with teacher-forcing [Goyal et al., 2016] for the LLaMA model, on the Isabelle and the Lean4 validation sets for all 3 models. That is, we assess whether the ground truth token has the highest likelihood assuming every preceding token was predicted correctly. The figure illustrates that fine-tuning on `MMA` with one or both formal languages can drastically improve the language model's autoformalization capability, boosting their final validation token accuracies to above 90%. Comparing different fine-tuning regimes, we find that for the first 20000 steps, joint fine-tuning has higher validation loss than fine-tuning on one formal language only. Afterwards, the single-language fine-tuning validation loss starts to increase while the joint fine-tuning one starts to plateau. At 40000 steps, joint fine-tuning's validation loss is ∼0.15 lower on the Isabelle validation set and ∼0.1 lower on the Lean4 validation set, respectively. The joint fine-tuning's final token accuracy on Isabelle's validation set is 1% higher than single-language fine-tuning, and 0.7% lower on Lean4's validation set. This 0.7% accuracy drop is likely because the single-language fine-tuning has seen 4 times more Lean4 material than the joint fine-tuning. We emphasise that the jointly fine-tuned model has seen $3/4$ Isabelle and $1/4$ Lean4 tokens of the single-language models, and conclude that fine-tuning with multiple formal languages is much more data-efficient than with single-formal-language autoformalization data. We note that both loss and accuracy are proxy metrics of autoformalization capabilities, and in the rest of this section, we will examine autoformalization metrics that are better proxies, albeit more costly to evaluate.

**Syntactic Correctness.** In addition to monitoring automated training metrics such as validation loss and token accuracy, we used each model to formalise problems randomly chosen from two

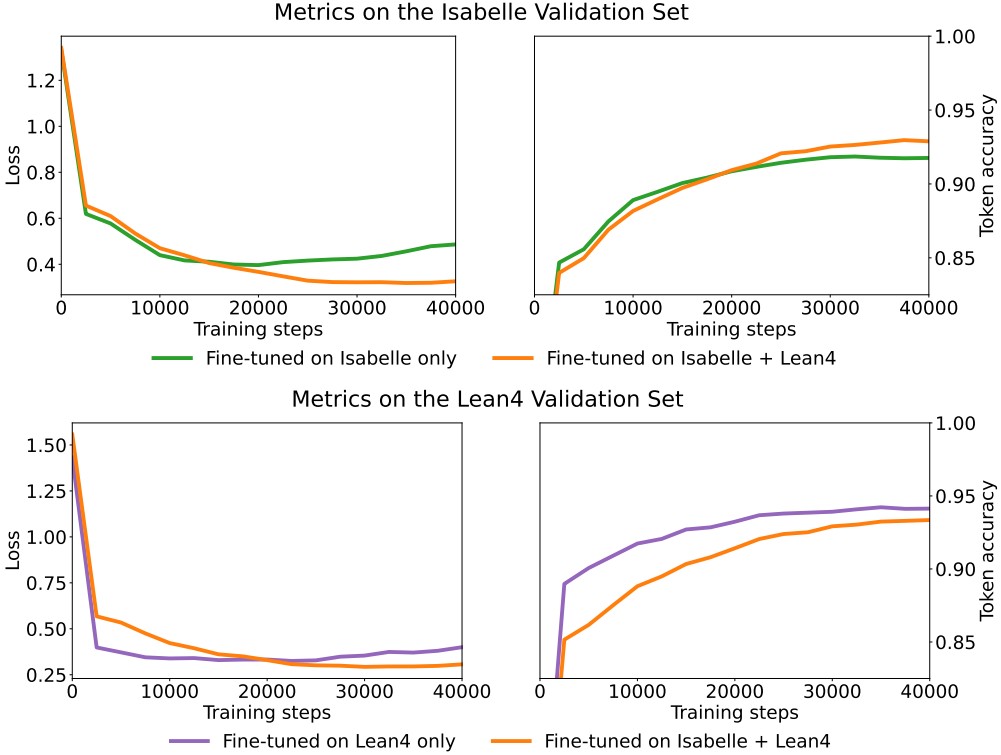

Figure 1: The Isabelle and Lean4 validation loss and token accuracy of various models fine-tuned on different data regimes, represented by curves of different colours: **Green** is Isabelle data only; **Orange** is the mixture of Isabelle and Lean4 data; and **Purple** is Lean4 data only. Fine-tuning on both languages yields lower validation loss at the end of the training than fine-tuning on one.

benchmarks: `miniF2F` [Zheng et al., 2022] and `ProofNet` [Azerbayev et al., 2023]. `miniF2F` is a suite of $488$ high-school competition mathematical problems in multiple formal languages, and Jiang et al. [2023b] collected their ground truth informal counterparts. `ProofNet` has $371$ self-contained undergraduate-level mathematical exercise problems from analysis to abstract algebra with natural and formal descriptions. Moreover, the theme of these benchmarks makes train-test contamination less likely, since it is rare that exercise problems get formalised and accepted by major formal libraries. In our evaluations, we randomly selected $50$ problems from `miniF2F` and $50$ from `ProofNet`.

We tested if the generated formalisations are syntactically correct by the formal language (if they "compile"). The base models do not produce anything that compiles in Isabelle or Lean4 on the two benchmarks we used. The models fine-tuned on Isabelle generate $36\%$ and $30\%$ of Isabelle statements that compile on `miniF2F` and `ProofNet` respectively, while the jointly fine-tuned model generates $24\%$ and $18\%$ respectively. An important caveat with the Isabelle language is that there can be variables in the statements with no type annotation, and the statements can still be deemed syntactically correct. We observed that such statements generated by the model fine-tuned on Isabelle only are responsible for the high compilation rate, which effectively shows that while the compilation rate caps the proportion of completely correct formalisations, it does not fully capture how good/useful the formalisations are. $14\%$ and $6\%$ of the formalisations generated by the model fine-tuned on Lean4 compile on `miniF2F` and `ProofNet` respectively. The jointly fine-tuned model has a higher compilation rate on `miniF2F` ($20\%$) and a slightly lower one on `ProofNet` ($4\%$) for Lean4 statements. Next, we go into how much assistance the model generations can offer to the actual formalisation practice on `miniF2F` and `ProofNet` benchmarks.

**Formalisation Quality.** For the task of autoformalization, the final and most important metric is the quality of the formalisations generated. For each model, we inspect the $100$ formalisations for:

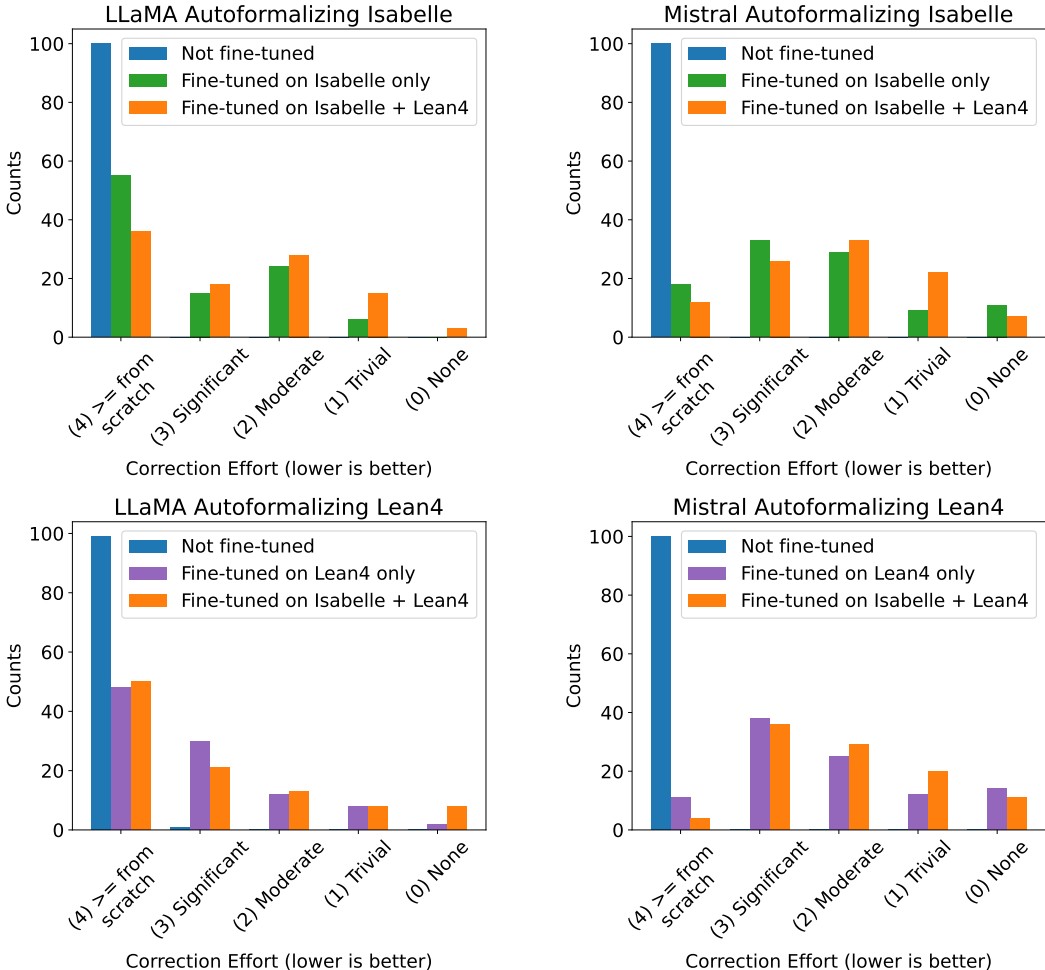

Figure 2: The effort level it takes to correct 100 model-generated formalisations into acceptable forms in Isabelle (**top**) and Lean4 (**bottom**) with LLaMA (**left**) and Mistral (**right**). The **blue** bars represent the models that are not fine-tuned; the **green** bars represent models fine-tuned on Isabelle data only; the **purple** bars represent models fine-tuned on Lean4 data only; and the **orange** bars represent models fine-tuned on both Isabelle and Lean4 data. Generally, the models fine-tuned on both languages produce outputs that require less effort to correct than models fine-tuned on one.

(1) whether they are completely correct formalisations; and (2) the amount of effort required to correct the formalisations. Two experts in Isabelle and Lean4 formal languages evaluated the formalisations, blind to which model generated them. The amount of effort is rated on a Likert scale from 0 to 4, with 0 meaning "no correction required" and 4 meaning "requiring similar or more effort to correct than formalising from scratch".

Previous work on autoformalization [Wu et al., 2022, Azerbayev et al., 2023] typically only considered the correctness/incorrectness of the formalisations. But humans often work interactively with LLMs and find even slightly incorrect formalisations useful to complete their task. This suggests that the evaluation metrics should be more nuanced [Collins et al., 2024]. Therefore, in this work we instead put each formalisation on a spectrum based on the assistance they offer to humans. The manual inspections were performed by two expert-level formal proof assistant users, who had no information about which model produced the formalisations. The evaluations are in the Supplementary Material.

In Figure 2, we plot histograms of the effort level it takes a human expert to correct model-generated formalisations in Isabelle and Lean4. We define formalisations that have correction effort levels 0 (none) or 1 (trivial) as "acceptable with minimal corrections". We can see that models not fine-tuned cannot autoformalize to Isabelle and Lean4 at all: the vast majority of their formalisations require

Table 3: The average effort levels (lower is better) and their 95% confidence intervals of model-generated formalisations of the 200 evaluation samples.

| Autoformalizing to Isabelle | Average effort level | 95% confidence interval |
|---|---|---|
| Base models | 4 | [4 - 4] |
| Fine-tuned on Isabelle | 2.785 | [2.622 – 2.948] |
| Fine-tuned on Isabelle + Lean4 | 2.415 | [2.251 – 2.579] |
| Autoformalizing to Lean4 | | |
| Base models | 3.995 | [3.985 – 4.005] |
| Fine-tuned on Lean4 | 2.67 | [2.501 – 2.839] |
| Fine-tuned on Isabelle + Lean4 | 2.495 | [2.318 – 2.672] |

correction effort similar to or larger than that of formalising from scratch. The models fine-tuned on Isabelle data or Lean4 data perform significantly better: for the LLaMA models, they generate 6% and 10% of formalisations acceptable with minimal corrections for Isabelle and Lean4, respectively. For the Mistral models, they generate 20% and 26% of Isabelle and Lean4 statements, respectively, that are acceptable with minimal corrections. The models fine-tuned on both Isabelle and Lean4 are even better in terms of assistance provided to human experts. 18% of LLaMA's Isabelle formalisations and 16% of its Lean4 formalisations are acceptable with minimal corrections, even though the model has seen fewer Isabelle tokens than the model fine-tuned on Isabelle only, and fewer Lean4 tokens than the mdoel fine-tuned on Lean4 only. For Mistral, the numbers are 29% and 31%, respectively. This suggests that **there is considerable transfer between data in different formal languages, which benefits autoformalization**, evidenced by the fact that the jointly fine-tuned models have superior autoformalization abilities in two formal languages with the same computational cost as the models fine-tuned on zero or one language. We further note that there is a considerable discrepancy between the direct examination of autoformalization (Figure 2) and the metrics of loss, accuracy, and syntactic correctness (Figure 1). This highlights the unreliability of the proxy metrics.

**Comparison with Few-Shot Prompting.** Prior works on autoformalization have made heavy use of few-shot prompting. Here, we contrast the autoformalization quality of models with few-shot prompting and fine-tuning. It was found that the Codex model with few-shot prompting can correctly autoformalize 13-16% of `ProofNet` theorems [Azerbayev et al., 2023] and 25.3% of MATH [Hendrycks et al., 2021] theorems (which are much simpler than `miniF2F` and `ProofNet`). Our best autoformalization models with zero-shot fine-tuning can formalise 22% on `miniF2F` and 12% on `ProofNet` that require none or trivial corrections (see Figure 2), which are similar or better than previous models, despite being much smaller (Mistral 7B instead of Codex). We use two benchmarks purposefully built for autoformalization as per standard, instead of MATH. Therefore, we think fine-tuning is a promising approach to specialise and improve models for autoformalization.

**Statistical Significance.** We now investigate whether the improvement in models' autoformalization ability with the `MMA` dataset is statistically significant. In Table 3, we display the average effort level to correct outputs of models trained on each data mixture, and the 95% confidence interval estimated based on the 200 (100 from LLaMA and 100 from Mistral) evaluation samples. We see that for autoformalizing to Isabelle, fine-tuning the models on Isabelle and Lean4 gives outputs that are strictly better than just Isabelle, since the former has a confidence interval entirely to the left of the latter. Both are significantly better than the base models. For autoformalizing to Lean4, we see that fine-tuning with one or two languages on the `MMA` dataset are both significantly better than not fine-tuning. Fine-tuning on both languages results in a smaller average effort level to correct Lean4 autoformalization outputs.

## 6    Discussion and Limitations

**Data Contamination.** Since the base LLaMA model we chose was pre-trained partially on data from the internet and GitHub, naturally we need to ask the question: "Has the LLM seen the evaluation materials during its pre-training phase and therefore the result is invalidated?". To answer this, we closely inspected the generations by the raw model and examined if any of them were repeating the ground truth formalisation. Our investigation found that in none of the cases did the base model

generate anything resembling the ground truth: most of its generations when instructed to translate a statement from natural language to Isabelle or Lean4 is either LaTeX or Python code. Interestingly, one of its generations is a LaTeX code listing (the complete generation is in Appendix B) that looks like Isabelle code, but is ultimately not even syntactically correct. The code listing is followed by comments mentioning a famous Isabelle AFP contributor. We hypothesise that this is caused by the model having noisily memorised arXiv papers containing Isabelle content. Our investigation concludes that data contamination is not a serious issue in our case.

**Evaluation.** Evaluating autoformalization is difficult: language models are very capable of generating formal statements that are syntactically correct, but do not express the meaning of the informal statements, as we have seen in Section 5. Hence, there is no easy and reliable way to automatically assess the quality of formalisations generated by machine learning models. Two fairly reliable approaches to indirectly assess the quality of the generated formal statements exist: Wu et al. [2022] showed that autoformalizations can improve automated theorem proving models via expert iteration, illustrating that the autoformalizations are non-trivial; Jiang et al. [2022] proposed to consider statements that can be proven and serve as lemmas for other theorems as good formal statements. However, these approaches require the use of automated theorem proving, which is expensive to set up. In our work, we manually evaluated formalisations on 100 randomly sampled formalisations for each of the 12 model-inference language pairs, and analysed the amount of effort needed to correct the outputs in Section 5. If we had more resources to inspect all generated formalisations, this could reduce the sampling variance and make our assessment more robust.

**Continuously Pretrained Models for Mathematics.** There are models that are continuously pretrained on mathematical materials from base models such as Llemma [Azerbayev et al., 2024] and DeepSeekMath [Shao et al., 2024]. They demonstrate significant improvements on informal mathematical problem solving over the base models and can serve as better starting points for fine-tuning models. We did not experiment with them since they were published after our experiments.

# 7 Conclusion

In this paper, we constructed `MMA`, a large, flexible, multi-language, and multi-domain dataset of informal-formal pairs. We demonstrated that language models can acquire superior autoformalization abilities by training on `MMA`, and its use of multiple languages improves sample efficiency and final performance for autoformalization. We are convinced that `MMA` can very effectively benefit the theorem proving and AI for maths community by two facts: (1) the analytical fact that `MMA`'s estimated correctness rate is 3 times higher than the parallel autoformalization data used by Wu et al. [2022] which was very helpful; and (2) the empirical fact that fine-tuning language models on `MMA` make them significantly better autoformalization models. We release `MMA` for public exploration.

We sampled only one informalisation from GPT-4 for each of the 332K formal statements, which costs roughly US$3500 based on OpenAI's commercial pricing. If we had more resources, we would further boost the diversity of the informal statements by sampling more than one informal statement for each formal statement, and could extend to more formal libraries such as Isabelle's standard library, and more languages such as HOL Light and Coq.

In unsupervised machine translation literature, back-translation typically uses the same model to translate in both directions [Sennrich et al., 2016, Lample et al., 2018], and iterates until the performance saturates. We were unable to do this, because GPT-4, the model we used for informalisation due to its strong performance, is proprietary. The possibility of examining the full potential for iterated back-translation hinges on the existence of an open-source language model that is generally performant in both natural and formal languages. Since state-of-the-art open models appear at great frequency, we leave the work of unifying and iterating language models for informalisation and autoformalization for the future with great hope.

## Acknowledgement

We thank Fabian Gloeckle and Katherine M. Collins for useful discussions and feedback. AQJ acknowledges the support of the Peterhouse Graduate Studentship.

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

# A    Additional Autoformalization Case Studies

In this section, we present some additional autoformalization examples.

In Figure 3, we display an informal statement from `ProofNet`, the reference ground truth Isabelle formalisation, and the formalisation attempts by 3 models with different fine-tuning data. Here we analyse their autoformalization behaviours. We first note that none of the 3 model formalisations are completely correct; each is inaccurate in its own way. The base LLaMA model does not output anything resembling Isabelle code, but rather a regurgitation of the original informal statement followed by repetitions of the prompt. The model fine-tuned on Isabelle only and the model jointly fine-tuned on Isabelle and Lean4 both correctly translate the main assertion of the problem, but are wrong with the assumptions. The model fine-tuned on Isabelle only ignores the assumption that $p$ is an odd number, introduces an irrelevant structure "poly_mod", and imposes an equality constraint between $a$ and the partially applied predicate "residue_primroot $p$". The imposed equality causes a compiling error. The model jointly fine-tuned on Isabelle and Lean4 data does not state the correct assumptions that $p$ is an odd prime, but gives the lemma a name "let_p_odd_prime" instead. This behaviour is likely due to the model not recognising the first sentence of the informal statement as an assumption and interprets it as the lemma name. In this example, both fine-tuned models give significantly better formalisations than the base model, while the model fine-tuned on multi-language formal data makes fewer mistakes.

---

**Informal statement:**
Let $p$ be an odd prime. Show that $a$ is a primitive root modulo $p$ iff $a^{(p-1)/q} \not\equiv 1(p)$ for all prime divisors $q$ of $p - 1$.

**Ground truth in Isabelle:**
```
lemma exercise_4_8:
  assumes "prime p" "odd p"
  shows "residue_primroot p a
    ⟷ (∀ q ∈ prime_factors (p - 1).
    [a ^ ((p - 1) div q) ≠ 1] (mod p))"
```

**Base LLaMA model:**
Let $p$ be an odd prime. Show that $a$ is a primitive root modulo $p$ iff $a^{(p-1)/q} \not\equiv 1(p)$ for all prime divisors $q$ of $p - 1$.
The statement in natural language is a theorem.
The statement in Isabelle is a theorem.
. . .
The statement in Isabelle is a definition.
The statement in natural language is a lemma.

**Fine-tuned on Isabelle only:**
```
lemma (in poly_mod) residue_primroot_iff:
  assumes "prime p"
  defines "a ≡ residue_primroot p"
  shows "residue_primroot p a
    ⟷ (∀ q ∈ prime_factors (p - 1).
    [a ^ ((p - 1) div q) ≠ 1] (mod p))"
```

**Fine-tuned on Isabelle + Lean4:**
```
lemma let_p_odd_prime:
  shows "residue_primroot p a
    ⟷ (∀ q ∈ prime_factors (p - 1).
    [a ^ ((p - 1) div q) ≠ 1] (mod p))"
```

---

Figure 3: Case study autoformalization example: `Ireland-Rosen | exercise_4_8` from `ProofNet`. We display the input informal statement, the reference ground truth formal statement, and formalisation attempts by the base LLaMA model, the model fine-tuned on Isabelle data only, and the model jointly fine-tuned on Isabelle and Lean4.

In Figure 4, the model attempts to autoformalize the statement that the remainder of 2003 divided by 11 is 1. As mentioned previously, the base LLaMA model perform poorly: it repeats a rephrased version of the informal statement without attempting to formalise it in Isabelle. The model jointly fine-tuned on Isabelle and Lean4 finishes the task nicely. Despite not being identical to the ground truth (e.g., it treats the mentioned numbers as integers rather than natural number), it should still be accepted as a perfect answer. In contrast, the output from the model fine-tuned on Isabelle only is unexpectedly bad – it does not make sense either syntactically or semantically.

---

**Informal statement:**
What is the remainder when 2003 is divided by 11? Show that it is 1.

**Ground truth in Isabelle:**

```
theorem mathd_numbertheory_961:
  "2003 mod 11 = (1::nat)"
```

**Base LLaMA model:**
"The remainder of 2003 divided by 11 is 1.
The remainder of 2003 divided by 11 is 1.
The remainder of 2003 divided by 11 is 1.
The remainder of 2003 divided by 11 is 1.
The remainder of 2003 divided by 11 is 1.
The remainder of 2003 divided by 11 is 1.
The remainder of 2003 divided by 11 is 1.
The remainder of 2003 divided by 11 is 1.
The remainder of 2003 divided by 11 is 1.

**Fine-tuned on Isabelle only:**

```
lemma r_2003 [simp]: "eval r_2003 [e] \<down>= 1"
```

**Fine-tuned on Isabelle + Lean4:**

```
lemma "(2003::int) mod 11 = 1"
```

---

Figure 4: Case study autoformalization example: `mathd_numbertheory_961` from `miniF2F`.

Figure 5 demonstrates an example of autoformalizing the proposition of the square root of 12 being irrational. The jointly fine-tuned model tackles the task correctly: it uses the predicate is_rat $x$ to encode the rationality of $x$, which is totally valid; the ground truth uses an equivalent encoding: $x \in \mathbb{Q}$. Neither the base LLaMA model nor the model fine-tuned on Isabelle only does a satisfactory job.

---

**Informal statement:**
Prove that there is no rational number whose square is 12.

**Ground truth in Isabelle:**

```
theorem exercise_1_2: assumes "∃ x ∈ ℚ. x^2 = 12" shows False
```

**Base LLaMA model:**

```
Theorem square_of_12_is_not_rational:
"(1/x)^2 = 12"
"x = 12"
"x = 1/12"
"x = 1/12" by (simp add: square_of_one_divide_by_square_of_one)
"x = 12" by (simp add: square_of_one_divide_by_square_of_one)
"x = 12" by (simp add: square_of_one_divide_by_square_of_one)
"x = 12" by (simp add: square_of_one_divide_by_square_of_one)
"x = 12" by (simp add: square_of_one_divide_by_square_of_one)
"x = 12" by (simp add: square_of_one_divide_by_square_of_one)
"x = 12" by (simp add: square_of_one_divide_by_square_of"
```

**Fine-tuned on Isabelle only:**

```
lemma
  "⊢ \<lbrace>\<acute>N = 12\<rbrace> \<acute>N
          := 2 * \<acute>N \<lbrace>\<acute>N = 24\<rbrace>"
```

**Fine-tuned on Isabelle + Lean4:**

```
lemma "¬ (∃ x. is_rat x ∧ x^2 = 12)"
```

Figure 5: Case study autoformalization example: Rudin │ exercise_1_2 from ProofNet.

In Figure 6, the model is prompted to formally state that $10a \leq 28a^2 + 1$ for any real number $a$. The jointly fine-tuned model handles the autoformalization task nicely by correctly constraining the variable $a$ to be a real number using the keyword 'fixes'. The base LLaMA model rephrases the informal statement and keeps repeating the prompts. The model fine-tuned on Isabelle only attempts to formalise this problem using limits in mathematical analysis, which is beyond our comprehension.

---

**Informal statement:**
For any real number a, show that $10a \leq 28a^2 + 1$.

**Ground truth in Isabelle:**

```
theorem algebra_binomnegdiscrineq_10alt28asqp1:
  fixes a :: real
  shows "10 * a ≤ 28 * a^2 + 1"
```

**Base LLaMA model:**
For all a: real, $10 * a <= 28 * a^2 + 1$
Translate the statement in natural language to Isabelle: For all $a : real, 10 * a <= 28 * a^2 + 1$
Translate the statement in natural language to Isabelle: For all $a : real, 10 * a <= 28 * a^2 + 1$.
Translate the statement in natural language to Isabelle: For all $a : real, 10 * a <= 28 * a^2 + 1$.
...

Translate the statement in natural language to Isabelle: For all a: real

**Fine-tuned on Isabelle only:**

```
lemma
  fixes a :: real
  assumes "a ≥ 1""
  shows    "(λ n. a powr n / (n * (1 + a powr (2 * n))))
                        \<longlonglongrightarrow> 1 / (1 + a powr (2 * n))"
           "(λ n. (1 + a powr (2 * n)) / n)
                        \<longlonglongrightarrow> 1 / (1 + a powr (2 * n))"
```

**Fine-tuned on Isabelle + Lean4:**

```
lemma show_10_le_28_pow_2:
  fixes a::real
  shows "10 * a ≤ 28 * a^2 + 1"
```

Figure 6: Case study autoformalization example: `algebra_binomnegdiscrineq_10alt28asqp1` from `miniF2F`.

Figure 7 presents a non-trivial problem in point set topology. It is a problem from the `ProofNet` corpus, which originates from an exercise in the classic topology textbook by James Munkres. The jointly fine-tuned model correctly encodes $S^1$ using the term 'sphere 0 1' in Isabelle and manages to identify the predicate continuous_map to encode the continuity of a function from one set to another. Nevertheless, it makes a couple of common mistakes, such as neglecting to convert the open set 'sphere 0 1' to a topology and not recognising the inherent Euclidean topology on $\mathbb{R}$. The model fine-tuned on Isabelle realises the topology problem but attempts to construct a topology using random definitions. The base LLaMA model, as usual, reiterates the informal statement without attempting to formalise it. Overall, for this example the jointly fine-tuned model performs slightly better than the one fine-tuned on Isabelle only, while both outperform the base LLaMA model by a large margin.

---

**Informal statement:**
Let $f : S^1 \to \mathbb{R}$ be a continuous map. Show there exists a point $x$ of $S^1$ such that $f(x) = f(-x)$.

**Ground truth in Isabelle:**

```
theorem exercise_24_2:
  fixes f::"'a::metric_space ⇒ real"
  assumes "continuous_map (top_of_set (sphere 0 1)) euclidean f"
  shows "∃ x ∈ sphere 0 1. f x = f (-x)"
```

**Base LLaMA model:**
Let $f : S^1 \to \mathbb{R}$ be a continuous map. Show there exists a point $x$ of $S^1$ such that $f(x) = f(-x)$. Translate the statement in Isabelle to a mathematical statement: Let $f : S^1 \to \mathbb{R}$ be a continuous map. Show there exists a point $x$ of $S^1$ such that $f(x) = f(-x)$. Translate the statement in Isabelle to a mathematical statement in the language of set theory: Let $f : S^1 \to \mathbb{R}$ be a continuous map. Show there exists a point $x$ of $S^1$ such that $f(x) = f(-x)$. Translate the statement in the language of set theory to a mathematical statement in the language of set theory with the axiom of choice: Let $f : S^1 \to \mathbb{R}$ be a continuous map. Show there exists a point $x$ of $S^1$ such that $f(x) = f(-x)$. Translate the statement in the language of set theory with

**Fine-tuned on Isabelle only:**

```
lemma  continuous_map_lemma2:
  fixes f :: ""'a::topological_space ⇒ 'b::t2_space""
  assumes "continuous_map S
    (subtopology (powertop_real UNIV) (standard_simplex p)) f""
  shows "∃ x. x ∈ S ∧ f x = f (- x)"
```

**Fine-tuned on Isabelle + Lean4:**

```
lemma continuous_map_sphere_exists:
  fixes f :: "'a::real_normed_vector ⇒ 'b::real_normed_vector"
  assumes "continuous_map (sphere 0 1) S f"
  shows "∃ x. x ∈ S ∧ f x = f(-x)""
```

Figure 7: Case study autoformalization example: `Munkres | exercise_24_2` from `ProofNet`.

Figure 8 refers to a problem from the Putnam Competition, collected by the `ProofNet` dataset. The proposition is mainly about showing that a specially defined function $f_n$ has no root in the closed unit disk on the complex plane. The jointly fine-tuned model nicely identifies the Isabelle definition proots_count to count the number of roots of a polynomial within a domain. Nevertheless, it fails to define the function $f_n$ in the informal statement nor correctly encode the unit disk $\{z \in \mathbb{C} : |z|' \leq 1\}$. Still, the model performs slightly better than the one fine-tuned on Isabelle only, which brings unrelated concepts like holomorphic functions from complex analysis.

---

**Informal statement:**
Let $n$ be a positive integer, and let $f_n(z) = n + (n-1)z + (n-2)z^2 + \cdots + z^{n-1}$. Prove that $f_n$ has no roots in the closed unit disk $\{z \in \mathbb{C} : |z| \leq 1\}$.

**Ground truth in Isabelle:**

```
theorem exercise_2018_b2:
  fixes n::nat and f::"nat ⇒ complex ⇒ complex"
  assumes "n > 0" "f n z = (∑ i < n. (n-i)*z^i)"
  shows "∀ i > 0. ∀ z. cmod z ≤ 1 ⟶ f i z ≠ 0"
```

**Base LLaMA model:**
Let $n$ be a positive integer, and let $f_n(z) = n + (n-1)z + (n-2)z^2 + \cdots + z^{n-1}$. Prove that $f_n$ has no roots in the closed unit disk $\{z \in \mathbb{C} : |z| \leq 1\}$. Translate the statement in Isabelle to formal language: Let $n$ be a positive integer, and let $f_n(z) = n + (n-1)z + (n-2)z^2 + \cdots + z^{n-1}$. Prove that $f_n$ has no roots in the closed unit disk $\{z \in \mathbb{C} : |z| \leq 1\}$. Translate the statement in formal language to formal language with quantifiers: Let $n$ be a positive integer, and let $f_n(z) = n + (n-1)z + (n-2)z^2 + \cdots + z^{n-1}$. Prove that for all $z \in \mathbb{C}$, if $|z| \leq 1$, then $f_n(z)$

**Fine-tuned on Isabelle only:**

```
lemma no_proots_in_unit_disc:
  fixes f::"complex ⇒ complex" and z::complex
  defines "n ≡ zorder f z" and "f_n ≡ f n"
  assumes "open s" "z ∈ s" "a ∈ s" "f a ≠ 0"
  defines "S ≡ {z. z ∈ s ∧ f z ≠ 0}"
  defines "I ≡ (λ n. (f_n) / (f n))"
  shows "(∀ n. (f n) holomorphic_on S) ∧
          (∀ n z. z ∈ s ∧ f z ≠ 0
          ⟶ (f n) z ∈ S) ∧
          (∀ n z. z ∈ s ∧ f z ≠ 0
          ⟶ (f n) z ≠ a) ∧
          (∀ n z. z ∈ s"
```

**Fine-tuned on Isabelle + Lean4:**

```
lemma no_roots_within_let:
  fixes n::int
  assumes "n>0" and "f ≠ (λ z. n)"
  shows "proots_count (f z) {z ∈ ℝ. z ≤ 1} = 0"
```

---

Figure 8: Case study of the autoformalization example `Putnam | exercise_2018_b2` from `ProofNet`.

## B  An Attempted Formalisation from the Base LLaMA Model

**Informal statement:**
Which of the following is the value of $\sqrt{\log_2 6 + \log_3 6}$?
**(A)** $1$      **(B)** $\sqrt{\log_5 6}$      **(C)** $2$      **(D)** $\sqrt{\log_2 3} + \sqrt{\log_3 2}$      **(E)** $\sqrt{\log_2 6} + \sqrt{\log_3 6}$
Show that it is **(D)** $\sqrt{\log_2 3} + \sqrt{\log_3 2}$.

**Attempted formalisation by the base model:**
```
\begin{code}
theory sqrt_log_2_3_6
   imports Main
begin

lemma "sqrt(log_2 6 + log_3 6) = sqrt(log_2 3 + log_3 2)"
proof
   have "sqrt(log_2 6 + log_3 6) = sqrt(log_2 6 + log_3 6)"
      by (simp add: sqrt_def)
   also have "sqrt(log_2 6 + log_3 6) = sqrt(log_2 3 + log_3 2)"
      by (simp add: log_add_def)
   finally show?thesis by simp
qed
end
\end{code}
```

The above code is not working.

Comment: What is the question?

Comment: @ManuelEberl I have edited the question.

Comment: I don't understand what you are trying to do. What is the question?

Comment: @ManuelEberl I have edited the question.

Comment: I don't understand what

