# OpenReview forum: "Multi-language Diversity Benefits Autoformalization"
_NeurIPS.cc/2024/Conference — NeurIPS 2024 poster_

### Official Review · Reviewer_8WKp · 2024-06-17

**Soundness:** 3
**Presentation:** 4
**Contribution:** 3
**Rating:** 7
**Confidence:** 4

**Summary:**

This paper introduces MMA, a large multi-language dataset for autoformalizing theorem statements. MMA employs a back-translation approach using GPT-4 to convert two formal corpora (AFP for Isabelle and mathlib for Lean4) into informal-formal pairs. Experiments demonstrate that LLMs such as LLaMA and Mistral, fine-tuned on MMA, achieve significant improvements over base models (from 0% to 29%-32% with no or minimal correction), outperforming their single-language counterparts in autoformalization on the miniF2F and ProofNet benchmarks.

**Strengths:**

* The paper is well-written and well-organized.
* The proposed MMA dataset is one of the largest collections of multi-language informal-formal data pairs, addressing the notable data scarcity in autoformalization. This dataset could be somehow beneficial for future research in this field.
* Experiments demonstrate that MMA is effective for fine-tuning LLMs for autoformalization, showing that multi-language data can enhance performance for single-language tasks.
* I appreciate the manual evaluation and the detailed discussion provided in this paper.

**Weaknesses:**

* I think the main weakness of this paper is that MMA is constructed solely based on zero-shot prompting for informalization by GPT-4, resulting in a dataset that is not perfectly aligned. Table 2 shows that the formalization accuracy is around 75%, indicating significant room for improvement. While the choice of zero-shot prompting may be due to cost considerations given the large scale of the dataset, there are potential ways to improve quality. For example, filtering out obvious errors in informalized statements or using more advanced prompting techniques (e.g., few-shot, self-consistency, or prompt to refine the misunderstanding concept) could help construct a smaller but higher-quality dataset. I am also curious about the effectiveness of fine-tuning with a small high-quality dataset compared to a large but noisy dataset for autoformalization.

* Minor: Missing references: There are some other works that leverage autoformalization to generate informal-formal data [1, 2]. [3] also surveys methods for autoformalization and theorem proving.

[1] FIMO: A Challenge Formal Dataset for Automated Theorem Proving, arXiv preprint 2023

[2] MUSTARD: Mastering Uniform Synthesis of Theorem and Proof Data, ICLR 2024

[3] A Survey on Deep Learning for Theorem Proving, arXiv preprint 2024

**Questions:**

* In the discussion, the author mentions plans to expand to other languages like Coq. However, I believe that a significant portion of Coq projects focuses on software verification, which is often challenging for human readability. One success of informalization by GPT-4 is that the formalized concepts are easily understood by GPT-4. Do you think your approach could still be effective for such Coq projects?

* (optional) If applicable, I am curious about the performance of DeepSeekMath or Llemma as the base model to be fine-tuned on MMA. These models have been pre-trained on large-scale informal and formal datasets and exhibit some basic autoformalization abilities.

**Limitations:**

I appreciate the detailed statement of the limitations in the paper. Moreover, I believe the most significant limitation is the presence of noisy aligned informal-formal pairs, which could be improved further.

---

> ### Author Rebuttal · Authors · 2024-08-06
>
> We would like to thank the reviewer for their detailed feedback and the time they invested in reviewing our paper. We address specific points raised by the reviewer:
>
> > **I think the main weakness of this paper is that MMA is constructed solely based on zero-shot prompting for informalization by GPT-4, resulting in a dataset that is not perfectly aligned. Table 2 shows that the formalization accuracy is around 75%, indicating significant room for improvement. While the choice of zero-shot prompting may be due to cost considerations given the large scale of the dataset, there are potential ways to improve quality. For example, filtering out obvious errors in informalized statements or using more advanced prompting techniques (e.g., few-shot, self-consistency, or prompt to refine the misunderstanding concept) could help construct a smaller but higher-quality dataset. I am also curious about the effectiveness of fine-tuning with a small high-quality dataset compared to a large but noisy dataset for autoformalization.**
>
> - We thank the reviewer for suggesting the advanced prompting techniques and filtering methods for improving the quality of the MMA dataset. In the paper, we explain that we did not use naive few-shot prompting because the formal data comes from many diverse domains, while few-shot prompting helps the most when the in-context examples are in the same domain as the input (line 156-157). We acknowledge that potentially one could write representative examples in each major category and dynamically retrieve them accordingly. However, considering that there are many diverse mathematical domains present in AFP and mathlib4, we think this approach is very labour-intensive and seems infeasible given our budget and access to formal mathematical expertise.
> - For filtering, as we explain in the informalisation taxonomy section, the informal outputs (even the wrong ones) make sense at first sight and require significant expertise to spot the mistakes. We cannot think of good ways to filter out major mistakes in the dataset at the moment.
>  -We gladly accept that there can be other methods to improve the alignment. As this is the first informalisation attempt at such a big scale, we are unsure how good the 75% accuracy is and hope for follow-up works to improve on it. The main reason for using a standard zero-shot prompting is that it provides the simplest methodology and as we are creating the dataset for the first time, it seems the most obvious choice. Improvements are welcome and can surely be made.
> - We would like to stress that the scope of this paper is to introduce a methodology and a dataset at a scale that has not been done before. More improvements are always possible, and this work will hopefully inspire and spark them. Based on these, we think the contributions made are significant enough.
>
> > **Minor: Missing references: There are some other works that leverage autoformalization to generate informal-formal data [1, 2]. [3] also surveys methods for autoformalization and theorem proving.
> [1] FIMO: A Challenge Formal Dataset for Automated Theorem Proving, arXiv preprint 2023
> [2] MUSTARD: Mastering Uniform Synthesis of Theorem and Proof Data, ICLR 2024
> [3] A Survey on Deep Learning for Theorem Proving, arXiv preprint 2024**
>
> We thank the reviewer for pointing out these works. Indeed, the autoformalizaton with feedback approach in FIMO and the joint theorem-proof autoformalization approach in MUSTARD are very nice additions for our context. We will update the paper by citing them.
>
> > **In the discussion, the author mentions plans to expand to other languages like Coq. However, I believe that a significant portion of Coq projects focuses on software verification, which is often challenging for human readability. One success of informalization by GPT-4 is that the formalized concepts are easily understood by GPT-4. Do you think your approach could still be effective for such Coq projects?**
>
> We think that the approach can still be effective for Coq projects. Our reason for this is that software verification often has a couple of important fixed themes like loop invariance, which allows for easier optimisation than mathematics. Furthermore, there are software and hardware verification theories in the AFP, which we did not find to be more difficult to informalise compared to mathemaitcal theories, from the informalisation taxonomy experiments.
>
> > **(optional) If applicable, I am curious about the performance of DeepSeekMath or Llemma as the base model to be fine-tuned on MMA. These models have been pre-trained on large-scale informal and formal datasets and exhibit some basic autoformalization abilities.**
>
> We thank the reviewer for pointing this out. Indeed DeepSeekMath, Llemma, or the recent Mathstral (continuous pretraining and fine-tuning based on Mistral 7B, so can be directly compared to Mistral 7B in the paper) can be very interesting subjects of study. As these models were not available when we started the paper, we did not venture to use them. We will update the paper to say that these additional training might make them better candidates for autoformalization models.
>
> We would like to thank the reviewer for their valuable feedback, which has greatly helped us improve our paper. Given the improvements we've made, we kindly request the reviewer to reconsider their score, or give indication for further improvements.

---

> > ### Comment · Reviewer_8WKp · 2024-08-09
> >
> > Thank you for your detailed response. I acknowledge that the MMA is the first large-scale approach to building an aligned informal and formal corpora, and I appreciate the manual efforts involved in the experiments. Your clarification has addressed my concerns, and I have revised my score positively in light of this.

---

> > > ### Author Response · Authors · 2024-08-11
> > >
> > > We want to thank the reviewer again for their valuable time and input. It has greatly helped us improve the paper. Thank you!

---

### Official Review · Reviewer_ByiA · 2024-07-06

**Soundness:** 3
**Presentation:** 3
**Contribution:** 2
**Rating:** 6
**Confidence:** 4

**Summary:**

The paper presents a dataset MMA and a model trained on the same. The dataset comprises informal-formal pairs of theorem which are generated using LLM. The experiment shows decent results on the autoformalization tasks of miniF2F and ProofNet benchmarks. The authors also claim that autoformalization performance improves when trained on multi-lingual data. These characteristics generalize over different sizes of models, supporting the hypothesis that multi-lingual training is good for autoformalization.

**Strengths:**

1. The authors present a strong case for how and why multi-lingual training is beneficial for autoformalization. This indicates the need for further studies on how transfer learning helps in training across different formal languages. We need to understand if it is only applicable to Lean and Isabelle or is applicable to other languages like Coq as well.


2. Adequate human studies have been conducted to test different aspects of the training data generated. The authors also try to thoroughly test the statistical significance of their findings which is great.


3. I appreciate the effort put in to manually verify a set of autoformalizations generated during the evaluations.

**Weaknesses:**

1. Even though, authors try manually to sample from the data, the sample size is often small (which is understandable). However, I want to see how the numbers change with varying sample sizes. For example, instead of just one experiment with 100 sample size, the same experiment with varying sample sizes of 50, 75, and 100 showing similar trends will be appreciated.

2. Authors make a key observation that informalization is much easier than formalization, however, I don't find enough justification for this in the paper. Some studies are mentioned on Page 4, but these observations may not generalize well on the dataset in question.

3. The authors accept that the GPT-4 informalizations should be considered as a noisy approximation. This actually is a big limitation. Since not every informalization is verified, there is a potential chance of leakage of the test data itself. Since informalization is generated by GPT-4, it might as well generate informalization for theorems other than the formal statement in question (maybe something which was similar to the formal statement). If such a similar formal statement is present in the test data, then this potentially can help the model perform better on test data (this can be considered a type of leak in the training data). I can understand that this may not be a common case. However, some case studies about the resemblance between the train and test data should be conducted.

**Questions:**

1. Can you separate Table 2 (Right) into various Error Types per language? Right now all the 200 MMA formalizations from the samples are presented together without the distinction between Lean vs Isabelle.

2. The authors talk about few-shot prompting vs instruction prompting on Page 4. I understand the budget reasons, however, I would like to see some case studies showing the qualitative differences between the two strategies (for a small subset of data) in this domain.

3. In Figure 2 (top), the bar plot for "Llama Autoformalization Isabelle" shows almost no correct formalization for the Isabelle-only model. I would like to the examples of those autoformalization that were possible because of the Isabelle + Lean model. A qualitative analysis of when this type of transfer is effective will be appreciated.

**Limitations:**

The authors do try to discuss some limitations, however, I would like to see more. I'm interested in seeing the resemblance of the test and train data since it was generated by GPT-4, and there is a potential chance of data leakage. Since the generation of informalization cannot be controlled, this risk always exists and hence just a comparison with the baseline is not sufficient.

---

> ### Author Rebuttal · Authors · 2024-08-06
>
> We would like to thank the reviewer for their detailed feedback and the time they invested in reviewing our paper. We address specific points raised by the reviewer:
>
> > **Even though, authors try manually to sample from the data, the sample size is often small (which is understandable). However, I want to see how the numbers change with varying sample sizes. For example, instead of just one experiment with 100 sample size, the same experiment with varying sample sizes of 50, 75, and 100 showing similar trends will be appreciated.**
>
> - We appreciate the reviewer for raising this point. We want to first clarify that each model is evaluated on 100 randomly selected examples, with 50 from miniF2F and 50 from ProofNet. Hence, we can randomly select 50, and 75 from these 100, to perform the required analysis. We do exactly this below:
> - We examine the autoformalization quality of the Mistral model fine-tuned on both Isabelle and Lean4, on 50, 75, and 100 random examples respectively. The plots are in Figures 1 and 2 in the general response. We can see that for both languages, the autoformalization quality proportions change very little as we go from 50 to 75 to 100 examples. This validates the sufficiency of having 100 examples as we can project that with more examples (since they are all randomly chosen), the autoformalization quality will not change dramatically, and conclusions drawn based on 100 samples will generalise.
>
> > **Authors make a key observation that informalization is much easier than formalization, however, I don't find enough justification for this in the paper. Some studies are mentioned on Page 4, but these observations may not generalize well on the dataset in question.**
>
> Analytically, informalisation does not have as stringent syntactical requirements as formalisation, so is more easily accepted. Empirically, Wu et al. and Azerbayev et al. both experimentally observed that informalisation is easier than formalisation on two datasets of different difficulty levels. Therefore, we have reasons to believe that the observation generalises to our dataset.
>
> > **The authors accept that the GPT-4 informalizations should be considered as a noisy approximation. This actually is a big limitation. Since not every informalization is verified, there is a potential ...**
>
> When we conducted the taxonomy of the informalisation, we noticed that GPT-4 always closely followed the input it is given and never generated something that is unrelated. In the 200 randomly chosen examples, GPT-4 always closely followed the formal statement. Since the MMA dataset is gathered from AFP and mathlib4, they are disjoint from the miniF2F and ProofNet benchmarks. This is because miniF2F and ProofNet benchmarks are deliberately prevented from becoming entries in AFP and mathlib4 to prevent contamination. Hence, we see no reason to particularly worry about benchmark contamination in our work. We will update the paper by attaching the 200 randomly chosen informalisation examples for the reader to examine as well.
>
> > **Can you separate Table 2 (Right) into various Error Types per language? Right now all the 200 MMA formalizations from the samples are presented together without the distinction between Lean vs Isabelle.**
>
> We appreciate the reviewer’s suggestion and present the error types per language below:
> | Error type    | Isabelle | Lean4 |
> | -------- | ------- | ------- |
> | None | 81 | 67 |
> | Hallucination | 2 | 6 |
> | Misunderstanding concept | 11 | 18 |
> | Incorrect assumption | 2 | 9 |
> | Incorrect conclusion | 2 | 6 |
> | Incorrect type | 4 | 8 |
> We will update the paper to use the per-language breakdown.
>
> > **The authors talk about few-shot prompting vs instruction prompting on Page 4. I understand the budget reasons, however, I would like to see some case studies showing the qualitative differences between the two strategies (for a small subset of data) in this domain.**
>
> We would like to clarify that the reason behind going for instruction prompting instead of few-shot prompting is not cost. As we explain on Page 4, there are a large number of mathematical domains in AFP and mathlib4, making it impractical to manually come up with few-shot prompts for every single domain. Hence, we consider the former strategy not eligible for constructing the MMA dataset. We can further conduct the qualitative case studies if the reviewer deems it necessary.
>
> > **In Figure 2 (top), the bar plot for "Llama Autoformalization Isabelle" shows almost no correct formalization for the Isabelle-only model. I would like to the examples of those autoformalization that were possible because of the Isabelle + Lean model. A qualitative analysis of when this type of transfer is effective will be appreciated.**
>
> - We thank the reviewer for this suggestion. We will have one case study in the rebuttal and give a further analysis of the autoformalizations only doable by jointly trained models in the updated paper.
> - Case study 1.
>   - Informal statement: “Find the remainder when $91145 + 91146 + 91147 + 91148$ is divided by 4. Show that it is 2.”
>   - Ground truth: "theorem mathd_numbertheory_640: "(91145+91146+91147+91148) mod 4 = (2::nat)"
>   - Jointly trained model autoformalization: lemma "(91145 + 91146 + 91147 + 91148) mod 4 = (2::nat)"
>   - Isabelle-only model autoformalization: "lemma assumes "x^4 + 34578*x^3 + 1996*x^2 + 688*x + 2 = 0" shows "(9114*x^6 + 334575*x^5 + 496*x^8 + 353*x^7 + 4*x + 2) / 4 = 2"
>   - One can see that the Isabelle-only model gives a degenerate solution while the jointly trained model manages to correctly formalise it.
>
> We would like to thank the reviewer for their valuable feedback, which has greatly helped us improve our paper. Given the improvements we've made, we kindly request the reviewer to reconsider their score, or give indication for further improvements.

---

> > ### Comment · Reviewer_ByiA · 2024-08-07
> >
> > I'm satisfied with the author's responses and their experiments. Hence, I have increase my score. I hope to see more nuanced examples showing the effectiveness of the transfer between languages.

---

> > > ### Author Response · Authors · 2024-08-08
> > >
> > > We thank the reviewer for such a swift response and appreciate their reconsidering the score!

---

### Official Review · Reviewer_zhuU · 2024-07-13

**Soundness:** 3
**Presentation:** 3
**Contribution:** 3
**Rating:** 6
**Confidence:** 4

**Summary:**

The authors use backtranslation to create a large (~332k samples) dataset of formal-informal statement pairs in the Lean4 and Isabelle formal proving languages: they take formal samples from a Lean4 and an Isabelle proof library, and ask GPT4 to restate them informally. They use this dataset to finetune two open source LLMs (Llama 33B and Mistral 7B) on the task of autoformalisation: converting informal statements to formal ones, in Lean4 and in Isabelle, comparing the result of this finetuning to the initial state of the LLMs and ablating on training with both languages as opposed to just the target languages.

The authors invest time and manually evaluate the performance of their trained models on 50 samples from each of two small benchmarks, proofnet and miniF2F (total 100 per evaluated model), to draw their conclusions.

For both languages, the fine tuned models outperform the base models on autoformalisation. For Isabelle, they find that training with the combined dataset (containing both Lean4 and Isabelle samples) is better than the relevant monolingual, for Lean4 the difference is slightly less clear.

I note the mixed dataset contains mostly Isabelle statements: 244k/332k, and personally wonder if this relates to the found results - the influence of this ratio is not explored. This makes sense however: evaluating the correctness of autoformalised statements is non trivial, and easily automated metrics such as training loss, or per-token accuracy in a teacher forcing setting, are not immediately indicative of correct formalisation.

**Strengths:**

1. Provide dataset that will be useful for fine tuning work on autoformalisation, and greater than previously available datasets
2. Method for generating dataset is straightforward: can be extended in future with additional languages and samples, or remade with better models (for better backtranslation)
3. Demonstrate the positive value of dataset for finetuning models for autoformalisation (and carry out the non trivial evaluation needed to do this)
4. Some investigation of whether it is better to use multi- or mono-lingual data for fine tuning autoformalisation models
5. Clearly written
6. While this is in the appendix - autoformalisation case studies are interesting for understanding what is happening. Similarly, in main paper, the analysis of 200 samples from the generated dataset (table 2) helps get a sense of its quality

**Weaknesses:**

1. Dataset is automatically generated by an LLM, and no filters were applied or considered. Naturally, some of the LLM's informalisations will be incorrect (manually inspecting 200 samples from the dataset, the authors find 52 with errors (table 2)). I wonder if further manual inspection would have raised any obvious failure cases that could be automatically flagged and filtered from the full dataset.
2. I would have liked an investigation of impact of the [ratio of different languages in the training set] on the [performance of the fine tuned model]. Specifically because the models fine tuned on the full dataset did better on Isabelle than when trained only on Isabelle, whereas for Lean this wasnt as clear, despite Isabelle having the larger dataset, I wonder if the issue for Lean was that it was 'drowned out' by the amount of Isabelle samples in the data.
3. I don't know how open NeurIPS specifically is to dataset rather than method contributions, and this seems mostly a dataset contribution. I do think it is a valuable one though.
4. The dataset is based on GPT4 backtranslations, and the evaluations are of open source models before and after fine tuning with these backtranslations. I wonder, is this actually just a distillation of GPT4 capacities into open models, or does this dataset actually advance autoformalisation? A comparison with GPT4 autoformalisation ability (eg just on proofnet and minif2f) would help settle this question.
5. Majority of improvement in token accuracy and loss of the models happens in first few k steps (fig 1). Would have been nice to do an evaluation of the models (ie on the benchmarks) at that stage, to see if remainder of steps is necessary
6. *Important!* Missing baselines/discussion: the question of whether fine tuning (as done in this work) is at all the correct approach for autoformalisation is not raised. Given that there is existing work on using few-shot learning for autoformalisation, the question is relevant and cannot be ignored. The few shot alternative should be properly discussed. At minimum, the existing reported results (Azerbayev at al, Wu et al) should be recalled and compared to in this paper (to the extent the comparison is possible, see next point). (Ideally, the models would have been evaluated in a few shot setting on the same benchmark-subset as used in the paper, though I recognise this is a labour intensive request.)
7. *Important!* Nonstandard benchmarking: The models in this paper are evaluated on the (mixed) subsets of two different benchmarks, proofnet and minif2f. This makes it hard to compare the results to those reported in other autoformalisation works, which use the benchmarks in whole and separately from each other. I hope the authors can at least update the report to list the disaggregated results on the two benchmarks. Independently, please list exactly which samples were taken from the two datasets for the combined one, to allow accurate comparisons to these results in the future. The use of only a small subset of the main benchmarks (50 samples each) also weakens the significance of the results.

**Questions:**

Clarifications, nits, comments:
1. Are the MMA statements disjoint from those of ProofNet and MiniF2F?
2. line 62 "strong" autoformalisation: subjective
3. section 2 "rule based informalisation" vs "symbolic informalisation" it is not clear to me if these are the same or different. If the same, choose one term, if different, clarify difference.
4. "Manual alignment" also unclear
5. line 86: appreciate the reference to specific case study, but would appreciate some lines on what "syntactically incorrect segments" are directly in this paper for completeness.
6. line 93: some lines on how there is a setting where there is a target-to-source model, but not the other way around, would be appreciated. Speaking of which: given all this dataset is based on GPT4 backtranslations: how does GPT4 perform at autoformalisation?
7. line 95: "usually, the back translation process is..." - would appreciate a/some sources on this!
8. line 99 what language is Azerbayev et al 2023 on?
9. line 116 "high quality": subjective, not clear by what measure this is said
10. line 119 "satisfies both of the criteria": did you check for diversity of the informal statements? how did you measure it?
11. line 142 do you mean "generation" (not curation) cost?
12. nice to have: line 148 does gpt4 add further licenses? what is the license of the combined dataset (is it just "both", or do they merge)?
13. table 2: what's that massive isabelle statement (24331 characters)? is it an anomaly? what happened there?
14. line 178: what is misunderstanding a concept - can you give a concrete example? similarly line 181 with "type".
15. line 191 vs lines 168 and 126: are formal expressions complete and precise or inherently ambiguous? soften/clarify statements to resolve this apparent contradiction
16. line 205: "train" -> "finetune"
17. line 208: what languages are miniF2F and ProofNet in? comes up again in line 250
18. nice to have: would be nice to see metrics on lean when fine tuned on just isabelle, and vice versa. correctness evaluation probably too labour intensive, but loss/token-accuracy curves maybe. but maybe they'll just be zero. i recognise this data may not be obtainable now that models have already been trained (don't know how hard/expensive it is to train the models)
19. line 227: number of epochs doesn't seem to quite line up with equalising number of samples seen/steps made, maybe because you are giving rounded numbers? doesn't particularly matter but strange
20. line 235: figure shows loss and per-token accuracy, which are not necessarily indicative of autoformalization capability, rephrase.
21. fig 1: not so clear what the per token accuracy of the models are in first 2k train steps, would appreciate knowing - maybe can have a line on this in body of paper if dont want to distort images
22. line 243: it has seen lean material 4x more, but it has not seen 4x more lean material (i.e. it is seeing same material)
23. line 242-243: unconvincing/invalid interpretation: if this were the explanation, we would see a similar (but weaker) drop for Isabelle, but instead, Isabelle sees a gain.
24. lines 245-246: lean4 results dont support this
25. line 258: "... generate 36% and 30% of Isabelle statements that compile..." unclear, rephrase
26. lines 256-269: seems compilation rate is discussed for only one model (Llama?). Would prefer to have information on both models. If not, at least clarify which model is being discussed here.
26. lines 262-265: need elaboration - I imagine you have support for this claim, but need to see it. Are you saying that a lot of the generated Isabelle statements type checked, but were incorrect? Give numbers
27. lines 265-267: more mixed results regarding whether it is better to train with mixed or monolingual data. I wish the paper was more up front on this inconclusivity, it appears to have a slight "joint-is-better" narrative, that is not critical to its contribution.
28. line 296: "mdoel"
29. line 308 again insufficiently up front about inconclusive results regarding mono/multi lingual data for lean4 (choosing to focus only on the comparison to not fine tuning at all)
30. line 309-310 don't particularly like this statement when the confidence intervals overlap (table 3), can be up front
31. need more of a gap under table 3
32. line 320: did fine tuning surface any memorised info? would have been interesting to check
33. line 340: call to "sample efficiency" - define and clarify how this relates to findings?
34. last 2 paragraphs of conclusion belong more in "limitations" in my opinion

35. what are the (math) domains of the data used to create MMA (within AFP and mathlib4), and the domains of miniF2F and proofnet, and how do they relate to each other? I.e., are miniF2F and proofnet in- or out-of- distribution for the MMA training set, with respect to type of math considered? What about specifically the subsets of miniF2F and proofnet that were used for the evaluations in this work?

**Limitations:**

yes

---

> ### Author Rebuttal · Authors · 2024-08-06
>
> We would like to thank the reviewer for their detailed feedback and the time they invested in reviewing our paper. We address specific points raised by the reviewer:
> > **Dataset is automatically ...**
>
> - We thank the reviewer for pointing out that there could be potential improvements to the dataset filtering process. We will include the complete informalisation evaluation results in the updated version for completeness.
> - We did not find any obvious failure cases that can be easily filtered out. Table 2 Right gives a breakdown of the errors made by GPT4 during the informalisation process. It can be seen that the most frequent errors are caused by the misunderstanding of formal concepts, e.g., not correctly understanding the context of inverse operations. We find that in general, spotting these errors requires a fair amount of domain expertise in mathematics and formal languages, and believe it to be difficult to find an automatic filter that does the same job.
> - We will update the paper to say this explicitly and let the readers judge after seeing the informalisations made.
>
> > **I would have liked an investigation of impact of ...**
>
> We agree with the reviewer that this is a valuable experiment to run. However, we are constrained by computational budget and unable to run the said experiments. To note, our main experiment with llama 30B took 14 hours on a TPUv4-64 to run, which by [TPU pricing](https://cloud.google.com/tpu/pricing) costs $2885. We are unable to do a sweep experiment that controls the Isabelle:Lean4 ratio and evaluate them accordingly. We recognise this as a limitation of the work and will update the paper to say so accordingly.
>
> > **I don't know how open ...**
>
> We think the methodology (back-translation + fine-tuning) can be expanded to other formal languages and datasets than Isabelle AFP and Lean4 mathlib4. The finding that multi-language diversity can make the resulting models stronger and more robust is a contribution in addition to the methodology. Hence we think the paper has more value beyond the dataset.
>
> > **The dataset is based on GPT4 backtranslations, ...**
>
> We think a direct comparison of our fine-tuned models with GPT4 has a confounding factor: model representation capability. The GPT4 model is rumoured to have 1.76T parameters while the models we fine-tuned have 30B and 7B parameters respectively. There is a 60-250x difference in model size and GPT4 thus has a significant advantage in terms of its representation capabilities.
>
> We think the fairest comparison would be to fine-tune a GPT4-level model on its own backtranslations, and see the model performance before and after the fine-tuning. It is impossible to do so for GPT4, but becomes more realistic as new open-weight models start to catch up and surpass GPT4’s performance. We leave this for future work as it still requires a significant amount of computational resources to adapt an open-weight model like Mistral Large 2 or Llama 3.1 405B for the autoformalization task.
>
> > **Majority of improvement ...**
>
> We performed some preliminary experiments on the intermediate checkpoints without performing the full suite of evaluation because it is expensive to do so. We find that the model’s autoformalization quality continues to improve as training progresses. We will update the paper to say so explicitly.
>
> > **Important! Missing baselines/discussion: ...**
>
> We agree with the reviewer that few-shot prompting should be compared as a competing technique. We will compare the results from Azerbayev et al. and Wu et al. below:
> - Azerbayev et al. (in Lean3): Codex with few-shot prompting can correctly autoformalize 13-16% of ProofNet theorems. ProofGPT 1.3B and 6.7B have 0 accuracies with few-shot prompting on the same dataset. ProofGPT-1.3B can correctly autoformalize 3.2% of ProofNet theorems after distilled backtranslation (similar to our methodology).
> - Wu et al. (in Isabelle): Codex with few-shot prompting can correctly autoformalize 38 out of 150 problems (25.3%) from MATH, which is of much lower difficulty than miniF2F and ProofNet.
> - For our best models without few-shot prompting, we can achieve 22% correctness on miniF2F, which is similar to the Wu et al. result but on a much harder dataset. We can achieve 12% correctness on ProofNet theorems, which is similar to Azerbayev et. al result but with a much much smaller model (Mistral 7B instead of Codex).
> - In view of the above comparisons, we think fine-tuning presents a very promising approach, which is compatible with few-shot prompting as well. But for simplicity, in this paper, we only consider the vanilla case of zero-shot fine-tuning. We will update the paper with these comparisons.
>
> > **Important! Nonstandard benchmarking: ...**
>
> - We note that only ProofNet (Azerbayev et al.) uses the complete dataset for comparison. The earlier work examining autoformalization (Wu et al.) used a subset of 150 MATH problems for evaluating autoformalization quality.
> - We will provide the disaggregated results on the two benchmarks. We already include the exact statements and evaluation results of the two subsets in the supplementary materials.
>
> We thank the reviewer for the very careful reading and feedback! Due to space limitations, here we will answer some questions. And for the rest of the points, we will modify the updated paper accordingly.
>
> 1. Yes
> 3. They are the same. We will update both to “symbolic informalisation”.
> 8. Lean3
> 10. Since the formal statements come from diverse mathematical fields, we expect the informal corresponding statements to also be diverse. We examined the informal statements and found it to be so.
> 12. GPT4 outputs are not constrained by licenses, only subject to Terms of Service by OpenAI.
>
> We would like to thank the reviewer for their valuable feedback, which has greatly helped us improve our paper. Given the improvements we've made, we kindly request the reviewer to reconsider their score, or give indication for further improvements.

---

> > ### Comment · Reviewer_zhuU · 2024-08-12
> > **Thank you**
> >
> > thank you for the updates! i will be glad indeed to see this paper updated with a discussion of & comparison (to extent that this is reasonably achievable) to few shot prompting, and disaggregated results on the benchmarks.
> >
> > ideally, i would also like results on the full benchmarks and not just subsets - regardless of whether other papers have also strayed in this way!

---

> > > ### Author Response · Authors · 2024-08-12
> > >
> > > We thank the reviewer for their response! We will do our best in making the comparisons and evaluations more compelling in the updated paper.

---

### Official Review · Reviewer_ctnj · 2024-07-17

**Soundness:** 2
**Presentation:** 3
**Contribution:** 3
**Rating:** 6
**Confidence:** 3

**Summary:**

This paper introduces MMA, a large-scale dataset consisting of informal-formal pairs of mathematical statements (i.e., parallel autoformalization data) in two types of formal languages, Isabelle and Lean4. The dataset encompasses statements from multiple domains and exhibits high quality. It was constructed using a back-translation method, converting from a formal language to an informal language, to improve data quality.

**Strengths:**

1.	The paper constructs a large-scale, high-quality, and diverse dataset for autoformalization.
2.	The paper analyzes the benefit of training on multiple formal languages to improve the performance of single-language autoformalization.
3.	The motivation for the method and the explanation for the data quality are written clearly.

**Weaknesses:**

The experiments and the corresponding analysis are not robust enough.
1.	The metrics “loss” and “token accuracy” used on the validation set are limited, as there could be multiple correct autoformalization results beyond the given reference. One piece of evidence for the limitation of these metrics is that in Figure 1, the model fine-tuned on Lean4 alone shows a higher loss, suggesting worse performance, yet its token accuracy is actually better.
2.	In line 242-243, the paper claims, “This 0.7% accuracy drop is likely because the single-language fine-tuning has seen 4 times more Lean4 material than the joint fine-tuning”. However, another potential explanation could be that the joint fine-tuning training data contains more Isabelle than Lean4, potentially leading to a deterioration in Lean4 performance. The authors should investigate this possibility.

**Questions:**

1.	Is it fair to control the number of training steps across different set of training data? Why not control for the size of the training dataset and the number of training epochs instead?
2.	In line 262-265, "it does not fully capture how good/useful the formalisations are", why are the formalizations considered not good/useful as long as they lack type annotations? What proportion of the statements are of this type? If these are removed, what is the proportion of syntactically correct statements generated by models that have been fine-tuned singly or jointly?

**Limitations:**

The metrics “loss” and “token accuracy” on the validation set are limited, as stated above. This could affect the correctness of all analyses and the corresponding conclusions.

---

> ### Author Rebuttal · Authors · 2024-08-06
>
> We would like to thank the reviewer for their detailed feedback and the time they invested in reviewing our paper. We address specific points raised by the reviewer:
>
> > **The experiments and the corresponding analysis are not robust enough.**
> > **1. The metrics “loss” and “token accuracy” used on the validation set are limited, as there could be multiple correct autoformalization results beyond the given reference. One piece of evidence for the limitation of these metrics is that in Figure 1, the model fine-tuned on Lean4 alone shows a higher loss, suggesting worse performance, yet its token accuracy is actually better.**
>
> We fully agree with the reviewer that the “loss” and “token accuracy” metrics do not show the full picture. We point out in line 270-271 that the final and most important metric for the task of autoformalization is that of the formalisation quality. For this metric, we measured it by having human evaluations of the difficulty/effort it takes to correct the machine-generated formalisations.
> We still wish to include the “loss” and “token accuracy” metrics for two reasons: (1) These two metrics are fairly standard in language model pre-training and fine-tuning; (2) We think by showing the discrepancy between these two metrics with the “ground truth” formalisation quality metric, we highlight their unreliability and emphasise the importance of the human evaluation. We will make the latter point more pronounced in the updated version of the paper.
>
>
> > **2. In line 242-243, the paper claims, “This 0.7% accuracy drop is likely because the single-language fine-tuning has seen 4 times more Lean4 material than the joint fine-tuning”. However, another potential explanation could be that the joint fine-tuning training data contains more Isabelle than Lean4, potentially leading to a deterioration in Lean4 performance. The authors should investigate this possibility.**
>
> We fully acknowledge and apologise for the lack of precision in our statement in line 242-243. We will update the paper to show both possible hypotheses: that the two fine-tuning runs have very different Lean4 data quantity, or that the two fine-tuning runs have very different Isabelle-Lean4 data proportions. Due to the limitation on computational resources, we are unfortunately unable to run a sweep of data proportion to confirm/refute the hypothesis, but have to leave it as future work.
>
>
> > **Is it fair to control the number of training steps across different set of training data? Why not control for the size of the training dataset and the number of training epochs instead?**
>
> This is a very good question! Controlling the total number of training steps and the number of training epochs have their respective advantages: controlling the number of training steps guarantees that the trained models go through an equal number of updates, and controlling the number of training epochs guarantees that the datapoints have been seen by the models for an equal number of times.
>
> In our preliminary experiments, models keep getting stronger as training progresses. In our current experiments, the joint fine-tuning consists of 3.3 epochs for both Isabelle and Lean4, while the Isabelle-only and the Lean4-only fine-tuning consist of 4.4 epochs and 13.2 epochs of the respective datasets (line 226-227). If we control the number of training epochs for the individual fine-tuning experiments, we will end up with training runs lasting 75% and 25% as long as current ones respectively. This will result in two significantly weaker models. Since the stronger models fine-tuned on single languages are worse than the jointly fine-tuned model, we have reasons to believe that their weaker versions are even worse. Therefore, this will not change our conclusion that multi-language training considerably benefits autoformalization capabilities.
>
>
> > **In line 262-265, "it does not fully capture how good/useful the formalisations are", why are the formalizations considered not good/useful as long as they lack type annotations? What proportion of the statements are of this type? If these are removed, what is the proportion of syntactically correct statements generated by models that have been fine-tuned singly or jointly?**
>
> As a design choice, open variables in theorem statements in Isabelle will be implicitly universally quantified: when properdivisor_sum is not a built-in definition, lemma "properdivisor_sum 18 = 21" is interpreted as lemma "!!properdivisor_sum. properdivisor_sum 18 = 21", where !! is meta-level forall. In contrast, in Lean4, variables need to be explicitly quantified within the statement (or declared earlier in the theory file). Due to this design discrepancy, the Isabelle syntax checker is more tolerant, allowing more false statements, whereas Lean’s syntax checker is stricter (at the cost of slightly more verbose statements).
>
> On a rough inspection (because inspecting very carefully takes as much time as re-doing the evaluations), we find that 19% of the Isabelle formalisations by the Llama model have this issue. So having this removed will reduce the syntactic correctness of the model fine-tuned on Isabelle alone to 14%, while the jointly fine-tuned model rarely has this issue and maintains a 21% syntactic correctness. In the updated version of the paper, we will do a closer inspection and report the corresponding figures.
>
> > **The metrics “loss” and “token accuracy” on the validation set are limited, as stated above. This could affect the correctness of all analyses and the corresponding conclusions.**
>
> We thank the reviewer for pointing this out! We will make it clearer that these two metrics have limited correlation with the model quality in the updated version, as discussed above.
>
> We would like to thank the reviewer for their valuable feedback, which has greatly helped us improve our paper. Given the improvements we've made, we kindly request the reviewer to reconsider their score, or give indication for further improvements.

---

### Author Rebuttal · Authors · 2024-08-06

We want to thank all reviewers for their keen input, which has significantly raised the quality of this paper. You will find a one-page attachment containing two plots we use to demonstrate our points.

Below, we'd like to address a few common points:

1. "Is the evaluation dataset size (100 per language) enough?" (reviewers zhuU and ByiA)
   - In the pdf attachment, Figures 1 and 2 show the distribution of correction difficulty for the jointly fine-tuned Mistral model for 50, 75, and 100 samples respectively. We can see that the distribution barely changes as the number of samples is increased.
   - This demonstrates empirically that the variance present in the 100 samples is small enough, and that adding more samples will not change the conclusion.
2. "Does advanced prompting work better?" (reviewers ByiA and 8WKp)
   - We did not use naive few-shot prompting because the formal data comes from many diverse domains, while few-shot prompting helps the most when the in-context examples are in the same domain as the input (line 156-157). Considering that there are many diverse mathematical domains present in AFP and mathlib4, we think few-shot prompting is very labour-intensive and seems infeasible given our budget and access to formal mathematical expertise.
   - We construct MMA as a dataset of this scale as a first step towards unlocking better autoformalization capabilities through fine-tuning. We want MMA to inspire and encourage further improvements, through better prompting techniques, while using our methodology.
   - We believe that our contributions are beyond the dataset itself, and warrant a publication.

Given the improvements and explanations we've made, we hope we can convince the reviewers to reconsider their score, or give indication for further improvements.

---

### Decision · Program_Chairs · 2024-09-25

**Decision:**

Accept (poster)

**Comment:**

This paper introduces MMA, a large-scale multi-language dataset for autoformalization, created using a back-translation approach with GPT-4. The dataset's quality and diversity enable improved performance in autoformalization tasks, particularly when training on multiple formal languages. The paper's strengths lie in its well-written presentation, valuable dataset contribution, and insightful experiments. However, there's room for improvement as suggested from reviewers (e.g., addressing the noise introduced by zero-shot prompting.) Overall, the paper provides a significant resource and insights for advancing the field of autoformalization.